

# Predictive value of immunoglobulin G, activated partial thromboplastin time, platelet, and indirect bilirubin for delayed viral clearance in patients infected with the Omicron variant

Lina Zhang[1,2,3,*], Shucai Xie[1,2,3,*], Feng Lyu[4], Chun Liu[5],
Chunhui Li[3,6], Wei Liu[1,2,3], Xinhua Ma[1,2,3], Jieyu Zhou[4], Xinyu Qian[4],
Yong Lu[7] and Zhaoxin Qian[1,2,3]

[1] Department of Critical Care Medicine, Xiangya Hospital, Central South University, Changsha, Hunan, China
[2] Hunan Provincial Clinical Research Center for Critical Care Medicine, Changsha, Hunan, China
[3] National Clinical Research Center for Geriatric Disorders (Xiangya Hospital), Xiangya Hospital, Central South University, Changsha, Hunan, China
[4] School of Computer Science and Engineering, Central South University, Changsha, Hunan, China
[5] Respiratory and Critical Care Medicine Department, The Third Xiangya Hospital of Central South University, Changsha, Hunan, China
[6] Xiangya Hospital, Central South University, Changsha, Hunan, China
[7] Department of Radiology, Ruijin Hospital Luwan Branch, School of Medicine, Shanghai Jiaotong University, Shanghai, China
* These authors contributed equally to this work.

Corresponding authors
Yong Lu, 18917762053@163.com
Zhaoxin Qian, xyqzx@csu.edu.cn

## ABSTRACT

**Background:** Omicron is the recently emerged highly transmissible severe acute respiratory syndrome coronavirus 2 variant that has caused a dramatic increase in coronavirus disease-2019 infection cases worldwide. This study was to investigate the association between demographic and laboratory findings, and the duration of Omicron viral clearance.

**Methods:** Approximately 278 Omicron cases at the Ruijin Hospital Luwan Branch, Shanghai Jiaotong University School of Medicine were retrospectively analyzed between August 11 and August 31, 2022. Demographic and laboratory data were also collected. The association between demographics, laboratory findings, and duration of Omicron viral clearance was analyzed using Pearson correlation analysis and univariate and multivariate logistic regression.

**Results:** Univariate logistic regression analyses showed that a prolonged viral clearance time was significantly associated with older age and lower immunoglobulin (Ig) G and platelet (PLT) levels. Using multinomial logistic regression analyses, direct bilirubin, IgG, activated partial thromboplastin time (APTT), and PLT were independent factors for longer viral shedding duration. The model combining direct bilirubin, IgG, APTT, and PLT identifies patients infected with Omicron whose viral clearance time was ≥7 days with 62.7% sensitivity and 83.4% specificity.

**Conclusion:** These findings suggest that direct bilirubin, IgG, PLT, and APTT are significant risk factors for a longer viral shedding duration in patients infected with

Omicron. Measuring levels of direct bilirubin, IgG, PLT, and APTT is advantageous to identify patients infected with Omicron with longer viral shedding duration.

## INTRODUCTION

Severe acute respiratory syndrome coronavirus 2 (SARS-CoV-2), a highly transmissible coronavirus originated in Wuhan City, China in late 2019, and caused a pandemic of acute respiratory disease (named 'coronavirus disease 2019', COVID-19), which is an emerging global health and public safety threat (*Umakanthan et al., 2020*). Since then, several variants of SARS-CoV-2, including D614G, Beta/Gamma, Delta, and Omicron, have been identified and catalyzed by four waves of the SARS-CoV-2 outbreak around the world (*Zhang et al., 2021*). As the most recent variant of SARS-CoV-2, the Omicron variant has caused a dramatic increase in COVID-19 cases worldwide since its discovery (*Setiabudi et al., 2022*). During the first week of January, 2022, more than 15 million positive cases of SARS-CoV-2 were reported in 149 countries.

Currently, the Omicron variant has become the dominant variant identified in patients with COVID-19. The Omicron variant has a very high risk of infection, as evidenced by its infection rates, which are four times higher than that of the wild type (*Araf et al., 2022*). A false-negative result in polymerase chain reaction tests further contributes to the spread of Omicron infection (*Torjesen, 2021*). More than 30 mutations have been identified in the conserved domain of the spike (S) protein of the Omicron variant (*Daria, Bhuiyan & Islam, 2022*). Accumulating evidence suggests that the heavily mutated Omicron variant S protein contributes to immune escape and weakens vaccine protection, ultimately leading to increased infectivity (*National Center for Immunization and Respiratory Diseases (NCIRD), Division of Viral Diseases, 2020*; *Shah & Woo, 2021*).

The Omicron variant of SARS-CoV-2 shows milder symptoms in patients, causes less severe disease, and has significantly fewer hospital admissions and deaths than past waves (*Christie, 2021*). However, based on its extreme adaptation and immune escape ability, Omicron and its subvariants may be dominantly prevalent for a period (*Zhang, Zhang & He, 2022*). In order to prevent the occurrence of unfortunate situations, all current treatments and management of SARS-CoV-2 are also necessary for Omicron variants (*Shuai et al., 2022*). Therefore, governments have to take the required steps to protect their countries from Omicron, including increased medical testing and screening, maintaining social distance, continuing vaccination for everyone, and isolating patients that test positive for the Omicron variant (*Araf et al., 2022*). At present, the recommended duration of home isolation in many countries is 5–7 days; however, a few patients have a time from diagnosis to virus shedding of >7 days. Viral shedding is a major consideration for patients to end isolation because a higher viral load (lower Ct values) means these patients are more contagious (*Li et al., 2022*). Thus, early identification of whether patients clear the virus in

a short period of time will contribute to therapeutic decisions, management measures, patient flow management, and resource allocation (*De Freitas et al., 2022*).

Previous studies have shown that prolonged SARS-CoV-2 shedding duration is associated with old age (*Li et al., 2022*), longer activated partial thromboplastin time (APTT) (*Yuan et al., 2021*), lower absolute lymphocyte (LYM) counts and lymphocyte-to-monocyte ratios (*Yuan et al., 2021*), and lower immunoglobulin (Ig)G levels in COVID-19 patients (*Li et al., 2020*). Laboratory findings are the early and easily identification of factors, which are useful for predicting hospitalization in omicron patients, especially in critically ill patients, and contributing to therapeutic decisions, and allocation of resources. A higher viral load and persistence were associated with a more severe disease course and higher in-hospital mortality rate (*Munker et al., 2021*). However, how laboratory findings relate to viral clearance is not yet understood in omicron patients. In the present study, 287 patients hospitalized with Omicron were analyzed and which hematological parameters associated with the duration of Omicron viral clearance were evaluated. Furthermore, a new predictive model capable of predicting patients infected with Omicron that had a virus clearance duration >7 days was constructed.

## MATERIALS AND METHODS

### Patients and data collection

Approximately 278 symptomatic patients were admitted to Ruijin Hospital Luwan Branch, Shanghai Jiaotong University School of Medicine from August 11, 2022, to August 31, 2022. Demographic and laboratory findings after admission were extracted from electronic medical records. All blood samples of omicron patients were included 3 days before the second negative detection of viral RNA. The laboratory findings of the blood sample were collected on the same day. If the patient has two blood samples with complete laboratory findings, the first one after admission was preferred. Peripheral blood specimens were prepared for complete blood count, routine hematological indicators of liver and kidney function, C-reactive protein (CRP), interleukin-6, procalcitonin, IgG, IgM, coagulation indicators (APTT, D-dimer, fibrin degradation products, fibrinogen, international normalized ratio, prothrombin time, and thrombin time), glucose, chloride, potassium, and sodium. Reverse transcriptase-polymerase chain reaction (RT-PCR) of oropharyngeal or nare (oropharyngeal/nare) swab sampling was used to determine if a patient was infected with Omicron and a Ct value <35 was defined as a positive result (*Viana et al., 2022*). Open reading frame (ORF) and nucleocapsid (N) genes of each patient were examined using RT-PCR every 24 h. A patient positive for Omicron infection was considered to be negative when the Ct value of both the ORF and N genes was higher than 35 for two consecutive tests.

### Study design

The duration of viral shedding was considered to be the interval between the blood test and the second negative detection of viral RNA. According to the duration of viral shedding time, the patients were divided into two groups: those that shed the virus for <7 days and those that shed the virus for >7 days. This study was approved by the ethics committee of

the Ruijin Hospital Luwan Branch, Shanghai Jiaotong University School of Medicine, Shanghai, China (No. 2022017). Because COVID-19 is a notifiable disease, individual patient consent was waived.

### Statistical analysis

The correlation between hematological parameters, age, and virus shedding duration was assessed by calculating the Pearson correlation coefficient. For continuous variables, the distribution was tested before analysis. Normally distributed continuous variables were displayed as the mean ± standard deviation (SD) and a *t*-test was used for comparison. Non-normally distributed continuous variables are shown as the median (25–75% interquartile range, IQR), and the Mann-Whitney *U* test was used for comparison. A Chi-square test was used to evaluate sex variables. Univariate and multivariate analyses were performed to analyze the association between hematological parameters and the duration of virus shedding. The performance of the factors and prediction models was evaluated using receiver operating characteristic curves (ROCs), and the area under the ROC (AUC), sensitivity, and specificity were determined. A *Z* test was performed to compare the predictive values of the different factors and models. *P* values < 0.05 were considered significant. All statistical analyses were performed using SPSS software (version 20.0; Chicago, IL, USA).

## RESULTS

### Pearson correlation of hematologic parameters and virus shedding duration

The correlation between hematological parameters, age, and virus shedding duration was assessed by calculating the Pearson correlation coefficient. As shown in Tables 1 and S1, age (r = 0.12, *P* = 0.029), direct bilirubin (Dbil, r = 0.10, *P* = 0.001), total bilirubin (Tbil, r = 0.14, *P* < 0.001), CRP (r = 0.14, *P* = 0.012), IgG (r = −0.29, *P* < 0.001), APTT (r = 0.13, *P* < 0.001), myoglobin (MB, r = 0.13, *P* < 0.001), LYM (r = −0.15, *P* < 0.001), platelet (PLT, r = −0.22, *P* < 0.001), white blood cell (WBC, r = −0.14, *P* < 0.001), creatinine (Cr, r = 0.08, *P* = 0.007), and neutrophil (Neu, r = −0.10, *P* < 0.001) levels had a relatively strong correlation with virus shedding duration. In the present study, age, Dbil, Tbil, CRP, serum amyloid A, APTT, MB, and Cr were significantly and positively correlated with the duration of viral clearance, while IgG, LYM, PLT, leukocyte, and Neu levels were negatively correlated with the duration of viral clearance. Therefore, these variables were included in subsequent analyses (Table 1).

### Comparison between demographic characteristics and hematologic parameters for different viral shedding duration groups

The demographic characteristics and hematological parameters of all patients are presented in Table 2. A total of 278 patients (123 males (123/278, 44.2%) and 155 females (155/278, 55.8%)) were included in the study. All patients cleared the viral infection, but two died (2/278, 0.7%). The age of the participants ranged from 19–100 years, with a median age of 73 years (25–75% IQR, 57–85 years). The median (IQR) of Dbil, Tbil, CRP,

**Table 1 Correlation between age, commonly used hematological parameters, and virus shedding duration (Pearson correlation).**

|  | Pearson's correlation coefficient | *P* value |
|---|---|---|
| Age | 0.12 | 0.029 |
| Dbil | 0.10 | 0.001 |
| Tbil | 0.14 | <0.001 |
| CRP | 0.14 | 0.012 |
| IgG | −0.29 | <0.001 |
| APTT | 0.13 | <0.001 |
| MB | 0.13 | <0.001 |
| Lymphocyte | −0.15 | <0.001 |
| PLT | −0.22 | <0.001 |
| WBC | −0.14 | <0.001 |
| Cr | 0.08 | 0.007 |
| Neu | −0.10 | <0.001 |

**Table 2 Demographic characteristics and laboratory findings of patients infected with Omicron (<7 day group and ≥7 day group).**

|  | All (*n* = 278) | <7 day group (*n* = 211) | ≥7 group (*n* = 67) | *P*-value |
|---|---|---|---|---|
| Age (IQR) | 73 (57–85) | 72 (55–83) | 79 (66–88) | 0.09 |
| Sex, M/F | 123/155 | 100/111 | 23/44 | 0.061 |
| Dbil, μmol/L (IQR) | 4.75 (2.8–8.11) | 4.96 (2.8–8.69) | 4.1 (2.4–6.69) | 0.088 |
| Tbil, μmol/L (IQR) | 10.22 (7–15) | 10.27 (7–15) | 10 (6.91–15) | 0.601 |
| CRP, mg/L (IQR) | 10.7 (2.6–59.66) | 10.23 (2.08–61.78) | 11.57 (3.91–58.95) | 0.597 |
| IgG (IQR) | 3.16 (0.45–28.24) | 4.92 (0.49–45.92) | 0.87 (0.31–7.46) | 0.02 |
| APTT, s (IQR) | 28.8 (26–32.58) | 28.3 (25.8–31.7) | 29.7 (29.1–33.7) | 0.029 |
| MB, ng/ml (IQR) | 62.05 (29.78–161.72) | 59.8 (24–156.3) | 74.33 (35.89–208.8) | 0.249 |
| Lymphocyte, ×10$^9$/L (IQR) | 1.1 (0.67–1.6) | 1.2 (0.71–1.65) | 0.9 (0.6–1.5) | 0.075 |
| PLT, ×10$^9$/L (IQR) | 185 (140.5–247.25) | 197 (147–261) | 156 (108–215) | <0.001 |
| WBC, ×10$^9$/L (IQR) | 6.28 (4.45–8.24) | 6.41 (4.51–8.23) | 6.16 (4.02–8.48) | 0.501 |
| Cr, μmol/L (IQR) | 74 (55–106) | 75 (55–106) | 71 (57–98) | 0.547 |
| Neu, ×10$^9$/L (IQR) | 4.2 (2.54–6.39) | 4.2 (2.59–6.2) | 4.01 (2.44–6.5) | 0.981 |
| Virus shedding durations (IQR) | 3 (2–6) | 2 (1–4) | 9 (7–12) | <0.001 |

IgG, APTT, MB, LYM, PLT, WBC, Cr, and Neu levels were 4.75 (2.8–8.11), 10.22 (7–15), 10.7 (2.6–59.66), 3.16 (0.45–28.24), 28.8 (26–32.58), 62.05 (29.78–161.72), 1.1 (0.67–1.6), 185 (140.5–247.25), 6.28 (4.45–8.24), 74 (55–106), and 4.2 (2.54–6.39), respectively (Table 2).

The Omicron group was divided into two groups based on viral shedding duration: <7 days (*n* = 211) and >7 days (*n* = 67). The results of the comparison between the two groups are presented in Table 2. Significant differences were observed in hematologic parameters between the two groups: IgG (*P* = 0.02), APTT (*P* = 0.029), and PLT (*P* < 0.001).

**Table 3** Univariate and multivariate logistic regression analyses of risk factors for viral shedding in patients infected with Omicron.

| | Univariable analysis | | | Multivariable analysis | | |
|---|---|---|---|---|---|---|
| | OR | 95%CI | *P*-value | OR | 95%CI | *P*-value |
| Age | 1.021 | [1.004–1.037] | 0.014 | 1.009 | [0.991–1.027] | 0.343 |
| Sex, M/F | 0.58 | [0.327–1.028] | 0.062 | 0.724 | [0.35–1.498] | 0.384 |
| Direct bilirubin | 0.954 | [0.901–1.011] | 0.111 | 0.806 | [0.692–0.938] | 0.005 |
| Total bilirubin | 0.982 | [0.951–1.014] | 0.271 | 1.087 | [0.988–1.195] | 0.087 |
| C-reactive protein | 1.000 | [0.995–1.004] | 0.939 | 0.999 | [0.991–1.006] | 0.767 |
| IgG | 0.993 | [0.987–0.999] | 0.017 | 0.993 | [0.987–0.999] | 0.033 |
| APTT | 1.032 | [0.998–1.078] | 0.154 | 1.098 | [1.026–1.175] | 0.007 |
| MB | 1.000 | [0.999–1.001] | 0.801 | 1.000 | [0.999–1.001] | 0.526 |
| Lymphocyte | 0.891 | [0.663–1.254] | 0.508 | 1.057 | [0.403–2.778] | 0.910 |
| PLT | 0.993 | [0.989–0.997] | <0.001 | 0.992 | [0.987–0.996] | <0.001 |
| Leukocyte | 1.020 | [0.953–1.092] | 0.559 | 1.014 | [0.420–2.448] | 0.976 |
| Cr | 1.000 | [0.999–1.001] | 0.795 | 1.000 | [0.998–1.001] | 0.673 |
| Neu | 1.029 | [0.962–1.101] | 0.406 | 1.11 | [0.451–2.729] | 0.821 |

## Univariate and multivariate analyses of associated factors that affect viral clearance of patients infected with Omicron

Univariate analyses were used to determine which factors were associated with a longer viral shedding duration in patients infected with Omicron. Older age (OR = 1.021; 95% CI [1.004–1.037]; *P* = 0.014) and lower IgG (OR = 0.993; 95% CI [0.987–0.999]; *P* = 0.017) and PLT (OR = 0.993; 95% CI [0.989–0.997]; *P* < 0.001) levels may significantly prolong the duration of viral shedding (≥7 days). However, based on multivariate analyses, Dbil (OR = 0.806; 95% CI [0.692–0.938]; *P* = 0.005), IgG (OR = 0.993; 95% CI [0.987–0.999]; *P* = 0.033), APTT (OR = 1.098; 95% CI [1.026–1.175]; *P* = 0.007), and PLT (OR = 0.992; 95% CI [0.987–0.996]; *P* < 0.001) were independent factors for longer viral shedding duration (≥7 days) (Table 3).

## Construction of a prediction model capable of predicting patients infected with Omicron that have a viral clearance duration >7 days

Using multivariate analyses, a new prediction model was constructed to distinguish whether the viral shedding duration of patients infected with Omicron was >7 days. The AUC of Dbil was 0.569, with the best cutoff point set at 9.455, and the sensitivity and specificity were 92.5% and 22.7%, respectively. The AUC of IgG was 0.627, with the best cut-off point set at 4.78, and the sensitivity and specificity were 74.6% and 50.2%, respectively. The AUC of PLT was 0.647, with the best cutoff point set at 161.5, and the sensitivity and specificity were 56.7% and 68.7%, respectively. The AUC of APTT was 0.588, with the best cut-off point set at 29.25, and the sensitivity and specificity were 59.7% and 57.3%, respectively (Table 4). In addition, the AUC value of the model combining Dbil, IgG, APTT, and PLT was 0.77, which was higher than those of Dbil (Z = 3.9343,

**Table 4 Predictive value of risk factors and the predictive model.**

|  | AUC | Sensitivity | Specificity | 95%Cl | Cut-off |
|---|---|---|---|---|---|
| Dbil | 0.569 | 0.92537 | 0.22749 | [0.493–0.646] | 9.455 |
| IgG | 0.627 | 0.74627 | 0.50237 | [0.555–0.699] | 4.78 |
| PLT | 0.647 | 0.56717 | 0.6872 | [0.574–0.720] | 161.5 |
| APTT | 0.588 | 0.59702 | 0.57346 | [0.513–0.664] | 29.25 |
| Model | 0.77 | 0.62687 | 0.83412 | [0.705–0.836] |  |

**Note:**
Model is composed of Dbil, IgG, PLT, APTT.

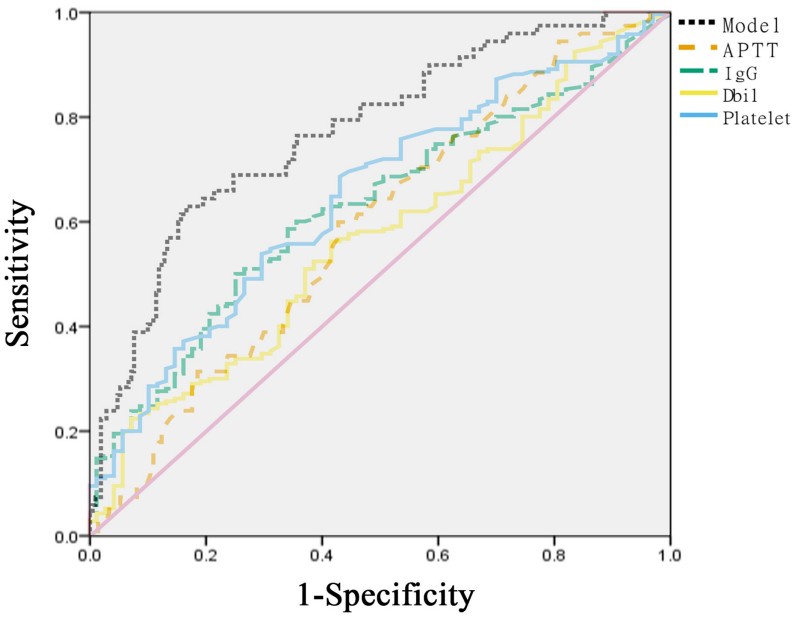

**Figure 1 Receiver operating characteristic curves of different risk factors and models.**

$P < 0.001$), IgG ($Z = 2.8843$, $P < 0.001$), APTT ($Z = 3.5624$, $P < 0.001$), and PLT ($Z = 2.4809$, $P < 0.001$) (Fig. 1).

## DISCUSSION

Omicron variants have become the predominant variant of SARS-CoV-2 worldwide, and the few studies on the duration of SARS-CoV-2 RNA shedding have generally been limited to an update of the isolation policy, which is based on SARS-CoV-2 variants that are no longer circulating. Prolonged SARS-CoV-2 shedding duration is associated with some laboratory findings. In the present study, age, direct bilirubin, total bilirubin, CRP, IgG, APTT, myoglobin, LYM, platelet, white blood cell, creatinine, and neutrophil levels were included in present study due to their relatively strong correlation with Omicron virus shedding duration. In addition, older age, Dbil, IgG, APTT, and PLT were potential risk factors affecting Omicron viral clearance.

Neutralizing antibodies are crucial for viral clearance in patients infected with SARS-CoV-2 *via* several mechanisms, such as interfering with virion binding to receptors,

blocking virus uptake into host cells, preventing uncoating of viral genomes in the endosome, or causing aggregation of virus particles (*Seow et al., 2020*; *Tang et al., 2021*). IgG antibody levels targeting the N, S protein, and receptor-binding domain are negatively correlated with viral load and related to viral clearance (*Ren et al., 2021*). Compared to that of the nonsymptomatic group, the symptomatic group have significantly lower virus-specific IgG levels in the acute phase and a significantly longer duration of viral shedding (*Long et al., 2020*). Mild COVID-19 groups may carry SARS-CoV-2 for a long time, which may be associated with the weak production of virus-specific IgG (*Guo et al., 2020*). Probiotic supplementation significantly increases specific IgG levels and improves symptomatic and viral clearance in outpatients with COVID-19 (*Gutierrez-Castrellon et al., 2022*). In addition, immunocompromised or patients with low humoral immune responses have longer virus shedding times (*Niyonkuru et al., 2021*; *Ye et al., 2020*). However, several studies have shown that patients with severe COVID-19 have higher concentrations of SARS-CoV-2-specific IgG than those with mild symptoms, but tend to have a high viral load and a longer virus-shedding period (*Liu et al., 2020b*; *Marklund et al., 2020*). Therefore, an Omicron-specific IgG antibody may be a critical risk factor affecting viral clearance.

Older age is a host factor affecting the immune system and underlying inflammation, which are major determinants of disease severity in COVID-19 (*Liu et al., 2020a*; *Lu et al., 2022*). IgG levels were significantly higher in older patients and in those with more severe disease, indicating that these patients have greater activation of their immune defense during recovery (*Li et al., 2020*). Compared to young patients, older patients usually have higher viral loads (*Chen et al., 2020*), a more severe course of COVID-19, and slower viral decline (*Li et al., 2022*; *Miller & Englund, 2020*). However, other studies have shown that median viral shedding is not significantly associated with age (*Fotouhi et al., 2021*). In this study, univariate analyses revealed that older age was a risk factor for delayed viral clearance. However, the results of multivariate analyses showed that older age was not an independent factor for longer duration of viral shedding. This finding was consistent with research results reported by *Lu et al. (2022)*. Among COVID-19 patients who were 60 years and older, PLT, DBIL, iBIL, and CRP were significantly higher than in younger patients who were less than 60 years old. Age may not directly related to the virus clearance of Omicron patients, but there is a strong correlation with IgG levels and laboratory findings. Therefore, further research is needed to confirm the effects of age on viral shedding duration. Prolonged viral shedding duration is associated with a longer APTT (*Yuan et al., 2021*), a lower LYM count, and a lower lymphocyte-to-monocyte ratio (*Li et al., 2020*; *Yuan et al., 2021*). A higher viral load and persistence are associated with a more severe disease course (*Munker et al., 2021*). Severe COVID-19 is accompanied by increased MB, CRP, Tbil, Dbil, Neu counts and decreased LYM counts, PLT, and Cr clearance rate (*Cao et al., 2020*; *Kronstein-Wiedemann et al., 2022*; *Lo et al., 2020*; *Lv et al., 2021*; *Zinellu et al., 2021*). In the current study, whether there was a correlation between these hematological parameters and Omicron viral clearance was determined. However, the result showed that the longer duration of Omicron was only associated with a longer APTT and lower PLT

levels. These inconsistent results might be due to the small sample size of this study, the characteristics of the selected population, and the differences in Omicron sub-lineages.

In the present study, a novel early prediction model combining DBIL, IgG, APTT, and PLT was established to distinguish longer viral shedding duration in patients infected with the Omicron variant. Determination of viral shedding duration will help reduce viral transmission, update isolation policies, make therapeutic decisions, and improve the scientific management of patients infected with Omicron. However, the applicability of the results in this study could be limited due to several factors. First, the number of samples and variables included in the model was small and validation of external data was lacking. Then, the population was from one geographical area in China; hence, these results may not be generalisable to other regions, subvariants of Omicron and populations. Further, information on the patients' medical history and current medical status could not systematically be included in present study. Therefore, a more efficient prediction model would require further research and development.

## CONCLUSIONS

In conclusion, Dbil, IgG, and PLT levels, as well as APTT were significant risk factors for longer viral shedding in patients infected with the Omicron variant and should be measured to identify those patients that have an increased viral shedding duration.

### Funding
Funding was provided by the National Key Research and Development Program of China (2022YFC2009800), Foundation of Shanghai Municipal Health Commission (202240204) and the China Postdoctoral Science Foundation (No.2022M713535) The funders had no role in study design, data collection and analysis, decision to publish, or preparation of the manuscript.

### Grant Disclosures
The following grant information was disclosed by the authors:
National Key Research and Development Program of China: 2022YFC2009800.
Foundation of Shanghai Municipal Health Commission: 202240204.
China Postdoctoral Science Foundation: 2022M713535.

### Competing Interests
The authors declare that they have no competing interests.

### Author Contributions
- Lina Zhang conceived and designed the experiments, performed the experiments, prepared figures and/or tables, authored or reviewed drafts of the article, and approved the final draft.
- Shucai Xie performed the experiments, analyzed the data, prepared figures and/or tables, authored or reviewed drafts of the article, and approved the final draft.

- Feng Lyu analyzed the data, authored or reviewed drafts of the article, and approved the final draft.
- Chun Liu conceived and designed the experiments, authored or reviewed drafts of the article, and approved the final draft.
- Chunhui Li conceived and designed the experiments, analyzed the data, prepared figures and/or tables, authored or reviewed drafts of the article, and approved the final draft.
- Wei Liu conceived and designed the experiments, prepared figures and/or tables, authored or reviewed drafts of the article, and approved the final draft.
- Xinhua Ma conceived and designed the experiments, authored or reviewed drafts of the article, and approved the final draft.
- Jieyu Zhou performed the experiments, analyzed the data, authored or reviewed drafts of the article, and approved the final draft.
- Xinyu Qian performed the experiments, analyzed the data, authored or reviewed drafts of the article, and approved the final draft.
- Yong Lu conceived and designed the experiments, authored or reviewed drafts of the article, and approved the final draft.
- Zhaoxin Qian conceived and designed the experiments, authored or reviewed drafts of the article, and approved the final draft.

## Ethics

The following information was supplied relating to ethical approvals (*i.e.*, approving body and any reference numbers):

The ethics committee of the Ruijin Hospital Luwan Branch, Shanghai Jiaotong University School of Medicine, Shanghai, China, approved the study (2022017).

## Data Availability

The raw measurements are available in the Supplemental File.

## Supplemental Information

Supplemental information for this article can be found online at http://dx.doi.org/10.7717/peerj.15443#supplemental-information.

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
