# Peer review of "Predictive value of immunoglobulin G, activated partial thromboplastin time, platelet, and indirect bilirubin for delayed viral clearance in patients infected with the Omicron variant"

_PeerJ, doi:10.7717/peerj.15443_

## Round 0.1 · original submission · Major Revisions

As stated by the reviewers, there are some underlying concerns that need to be addressed in terms of the model development and how further validations will be carried out. Please address these comments in the manuscript for its resubmission.

Reviewer 1 ·

Basic reporting

Overall the sentences are comprehensive and clear. However, certain lines are very broad and needs correction to sound specific. For example, in line 70, the sentence starting "However, to..." is vague. Also, in line 78, is there a missing reference at (P)?

The references and the background provided are adequate. However, the background still needs to explain the importance of understanding the correlation between laboratory findings and viral shedding in line 91.

Experimental design

Lines 88-91 says that the previous studies already showed the correlation between viral shedding and other factors. The sentence in Line 91 doesn't strongly justify the need to study the correlation again.

The experimental design needs to mention the time of blood collection from these patients, as the antibody response is different at different time points.
The vaccination and previous infection (i.e., Delta variant) history of patients is crucial to find variation in the patient demography.
Among the patients categorized as longer viral shedding (>7 days), how many were asymptomatic and symptomatic?

Validity of the findings

The results section (from line 137-157) is simply descriptive of the results obtained by the authors, but is lacking any/clear interpretation and conclusion pertaining to the data.
I see concluding sentences only in lines 163, and 167

In Table 1., the p values show significance, but the Pearsons correlation coefficient is less than 0.29 suggesting mild correlation. Maybe splitting the patient groups into symptomatic and asymptomatic can help increase the correlation. Also, the authors should comment on the negative correlation coefficient.
Lines 186-188 should go in the results section instead of the discussion section.
Given the low correlation coefficient, Line 189 should say "potential risk factor" instead of "significant".

The univariate and multivariate analyses seem meaningful and the interpretation could be more specific.

Lines 224-226 is vague, and needs to be revised to make clear conclusion.

The predictive model graph in Figure 1., shows better specificity compared to the individual parameters. This model can be validated easily by collecting blood samples from new omicron patients and determine their viral shedding based on their Dbil, IgG, platelet and APTT measure. doing this experiment could strengthen this finding.
In line 230 the authors say that the model have a better efficiency, but they cannot say that without validating the model as discussed above.

Additional comments

The commend the authors for their laborious data collection and analyses. Given the nature of human research, I understand to expect variation and lesser correlation with the given sample size. However, I believe some revisions are necessary to accept the authors claims.

Reviewer 2 ·

Basic reporting

In this study, the authors have investigated the association between Omicron viral shedding and host factors such as bilirubin, IgG, PLT and APTT. This work is well-written, however there are few concerns in the experimental design, data collection and rationale for the choice of variables.

Experimental design

The authors have examined 278 patients. Though the sample size is relatively small, the authors have tried to develop a prediction model to understand the host factors involved in omicron clearance. However, further validation of the model is required. This study could serve as a foundation for future research. A few of the concerns which needs to be addressed by the authors are:

1. There are different subvariants of Omicron that are in circulation. Did the authors categorize patients in relevance to the variants of Omicron strain? Additional data has to be included or mentioned as limitation of the study.

2. When the data for bilirubin, IgG, PLT and APTT were collected from patients, what were the medications that the patients were on? If the patients were on any antiviral treatment, the results of this study could vary substantially. Clarification regarding the antivirals the patients were on would improve the overall understanding. Please provide the data if available.

3. The rationale for choosing the parameters included in this study is not clear. How did the authors select the variables used in this study? Other variables which could be considered are vaccine, probiotic/vitamin intake, usage of glucocorticoids, long-term medication, underlying chronic conditions, etc. The authors should include an explanation for the choice of variables used in this study, preferably in the discussion section.

4. There are inconsistencies with respect to the correlation with age and APTT (Table 3) in univariable and multivariable analysis. This must be clarified.

Validity of the findings

Impact or validity of the prediction model have not been assessed.

---

## Round 0.2 · accepted · Accept

Though validation of the model was not carried out in this study, the model shows potential and the authors have highlighted the drawbacks of the study. This would help researchers in the field to use a similar model for validation in the future.

Reviewer 1 ·

Basic reporting

No comments

Experimental design

“Blood collected 3days before second negative detection of viral RNA”

This in my understanding means the blood was collected between the first and the second negative test.

It is still unclear if the blood was collected certain days post symptom onset or after the first positive test for viral RNA.

Validity of the findings

In the current form, the article still doesn’t have strong validation.

As the authors elegantly explain in lines 245-259, there are several factors influencing the poor correlation observed. Therefore without the further validation that the authors are claiming to be working on, this current manuscript is still lacking.

If the previous studies have observed this correlation and validated their findings in other SARS-CoV variants, that can be used to predict the impact of the authors findings.

Reviewer 2 ·

Basic reporting

The authors have now included the limitations of the study which helps in understanding the overall impact of their work. No further improvement is needed.

Experimental design

No comment

Validity of the findings

The authors have not validated the model. Future work is needed with increased number of samples from different geographical location.